# Wearable Prophylaxis Tool for AI-Driven Identification of Early Warning Patterns of Pressure Ulcers

**DOI:** 10.3390/bioengineering10101125

**Published:** 2023-09-25

**Authors:** Lorenz Gruenerbel, Ferdinand Heinrich, Jonathan Böhlhoff-Martin, Lynn Röper, Hans-Günther Machens, Arthur Gruenerbel, Moritz Schillinger, Andreas Kist, Franz Wenninger, Martin Richter, Leonard Steinbacher

**Affiliations:** 1Fraunhofer Institute for Electronic Microsystems and Solid State Technologies EMFT, 80686 Munich, Germany; franz.wenninger@emft.fraunhofer.de (F.W.); martin.richter@emft.fraunhofer.de (M.R.); 2Department for Plastic Surgery and Hand Surgery, Technical University Munich, Hospital Rechts der Isar MRI, 81675 Munich, Germanyleonard.steinbacher@mri.tum.de (L.S.); 3Bavarian Foot Network, 81477 Munich, Germany; gruenerbel@diabeteszentrum-muenchen-sued.de; 4Artificial Intelligence in Communication Disorders, Friedrich-Alexander-University Erlangen-Nürnberg, 91054 Erlangen, Germanyandreas.kist@fau.de (A.K.)

**Keywords:** pressure ulcers, decubitus, bedsore, foot ulcer, machine learning, digital health, AI, prevention, personalized medicine, continuous monitoring

## Abstract

As today’s society ages, age-related diseases become more frequent. One very common but yet preventable disease is the development of pressure ulcers (PUs). PUs can occur if tissue is exposed to a long-lasting pressure load, e.g., lying on tissue without turning. The cure of PUs requires intensive care, especially for the elderly or people with preexisting conditions whose tissue needs longer healing times. The consequences are heavy suffering for the patient and extreme costs for the health care system. To avoid these consequences, our objective is to develop a pressure ulcer prophylaxis device. For that, we built a new sensor system able to monitor the pressure load and tissue vital signs in immediate local proximity at patient’s predilection sites. In the clinical study, we found several indicators showing correlations between tissue perfusion and the risk of PU development, including strongly reduced SpO${}_{2}$ levels in body tissue prior to a diagnosed PU. Finally, we propose a prophylaxis system that allows for the prediction of PU developments in early stages before they become visible. This work is the first step in generating an effective system to warn patients or caregivers about developing PUs and taking appropriate preventative measures. Widespread application could reduce patient suffering and lead to substantial cost savings.

## 1. Introduction

In an aging society, the prevalence of age-related diseases rises. Pressure ulcers, also known as bedsores, pressure injury, or decubitus, are one of these illnesses that already have a significant negative impact on the healthcare system. To increase readability, the acronym for pressure ulcers “PU” will be used throughout this work. A PU is defined as a local tissue or skin damage that occurs as a consequence of applied pressure with or without additional shear stress. Mainly, PUs emerge at predilection sites such as bony prominences or during long-term exposure to hard materials. The consequences can reach from minor skin or tissue damages to large and very painful wounds. PUs rarely occur in patients with normal sensory function and mobility, since conscious and unconscious feedback leads to position shifts before irreversible tissue damage occurs. Many risk factors of PU development are known and tend to arise with age. Major risk factors include age, sensory impairment (e.g., neuropathy as a consequence of diabetes) or immobility, for example after surgery [1].

The main goals of this study are (1) to lower the pain experienced by patients and (2) to reduce healthcare costs. For that, prophylaxis tools are the most efficient way since they reduce the need for extensive care. Within the KIPRODE (German acronym for “AI for the prophylaxis of decubitus”) project [2], we developed a concept to build a prophylaxis system as shown in Figure 1. At the core of this concept a wearable sensor system monitors the pressure load applied on the skin as well as vital parameters of the skin at a predilection site. In order to correlate the sensor data with the skin’s health condition, a doctor’s assessment must also be recorded. Then, machine learning is used to find and evaluate an early warning pattern for PUs. The final application should operate on a smartphone or a wearable device and warn the patient or caregiver if the risk of PUs rises. This allows for a timely reaction and avoids the severe consequences of a PU. Especially in ambulatory settings, where regular inspection of predilection sites and turning or repositioning of the patient cannot be guaranteed, the prophylaxis tool would be of great use. Our partner MONKS [3] provides the required infrastructure for this work as well as future implementations with a privacy-secured online tool, that also allows us to run the data analysis algorithms.

### 1.1. Pressure Ulcers Prevalence and Health Care Cost

According to official statistics [4], there were almost five million people in need of care in Germany in 2021, of whom around 248,000 developed a severe PU (at least grade two) in the same year [5]. The global incidence of PUs is estimated to be around 3,200,000 in 2019 [6]. Due to the reduced healing ability of older patients and the extensive treatment required to heal pressure ulcers, the disease poses a major health risk to many individuals.

Additionally, pressure ulcers also lead to a significant economic burden. Even though only a small amount of data about actual prevention costs exists, Demarre et al. [7] compared several studies on PU prevention in 2015. According to them, the cost for preventive measures per patient and day varied between €2.65 and €87.57 across different institutions resulting in enormous expenses if the patients needed to be covered over long time periods. If prevention fails and a PU needs to be treated, the costs per patient per day can amount up to €470.49.

Anthony et al. [8] estimated the related health care cost by citing the cost of wounds (PUs are a part of the cost) in the UK, which amounts up to £5.1bn annually; the treatment cost of a pressure ulcer of the most severe stage in the US is estimated to be over US$129,000.

### 1.2. Pathogenesis of Pressure Ulcers

Pressure ulcers (PUs) are one of the most critical diseases that bedridden individuals and those with limited mobility can suffer from. The most common predilection sites for PUs are body areas with little soft tissue between the bone and the skin. Applying pressure to these areas over an extended period can lead to the formation of a pressure ulcer. Some of these sites include the skin over the sacrum, coccyx, hips, and shoulder blades, as well as the skin on the heels, elbows, knees, ankles, and the back of the skull [9]. The basic mechanisms involved in the development of pressure ulcers are directly influenced by the pressure or shear force on the skin and underlying tissues [10]. In most cases, the pressure is generated by the individual’s own body weight or by a medical device in contact with the skin. The initial pathology in response to any form of pressure is the reduced or interrupted blood supply to the affected area, causing hypoxia, i.e., reduced oxygen supply of the tissue. Prolonged pressure load and hypoxia can then cause inflammatory edema, which increases the pressure on the surrounding tissue and further impairs blood supply [10,11]. These mechanisms lead to ischemic damage to the skin and underlying tissue layers. This damage increases exponentially with the duration of this condition, leading to larger and deeper tissue defects [11].

The risk of developing a pressure ulcer depends on numerous factors that can affect the skin’s resistance to pressure. Some of these factors are a humid microclimate, atherosclerosis, paralysis, neuropathy (e.g., diabetes-related), malnutrition, increased age, BMI > 23, and hypertension [1,12,13,14].

PUs have a high risk for recurrence. Patients who already had a PU with or without surgery show a recurrence risk of up to 80% within the first years after treatment [15]. Very severe wounds may need to be operated and covered with new skin as the wound may not heal itself.

### 1.3. Pressure Ulcer Classification

The classification of pressure ulcers is commonly guided by the severity of tissue damage and categorized into four stages following the guidelines set by the European Pressure Ulcer Advisory Panel [16], the National Pressure Injury Advisory Panel [17], and the Pan Pacific Pressure Injury Alliance [18]. These stages range from simple skin erosion to deep muscular tissue damage, each requiring different treatment [9,19]. A stage one pressure ulcer describes a redness of the affected skin (hyperemia) that cannot be reduced or dissolved by pressure. This patho-mechanism is utilized to diagnose stage one pressure ulcers using the Phillips finger test.

For the finger test according to Phillips, a professional presses a finger to compress the red (hyperemic) skin of a patient and examines the spot where the finger is pressed. If there is no visual indication for a reduction of the hyperemia after the finger is removed, the Phillips finger test is considered positive, indicating a stage one PU. A stage two pressure ulcer is described as partial destruction of the skin in the pressure-exposed area that can reach down to the dermis. Commonly a flat, light red sore bed without scabs or layering is visible through these lesions. To be classified as a stage three PU, the lesion must be deep enough to clearly expose subcutaneous fat, while not revealing any visible bone, tendons, or muscle. A stage four pressure ulcer is only diagnosed when the affected area has severe tissue damage, showing the underlying bone, muscle, and tendon structures. Scabs and plaque are often present in these wounds as well.

### 1.4. Current Approaches for Pressure Ulcer Prevention

Regular inspection of predilection sites and turning or repositioning of the patient are clinical standards for preventing pressure injuries in hospitals. The literature frequently mentions turning intervals of two hours, but this guideline appears to be based more on economical considerations than on scientific findings [1,20]. To identify high-risk patients, hospitals often use risk assessment scales such as the Braden scale to implement suitable prevention programs [21].

Lechner et al. [22] summarize the outcome of 332 studies about the prevention of PUs. They conclude that several preventive measures exist that reduce pressure exposure on predilection sites. However, they also point out the lack of PU prevention methods with high accuracy, do not mention personalized wearable monitoring systems and state the need for further research. Furthermore, they mention the heterogeneity of study outcomes, which consequently makes comparisons difficult [22]. A possible solution for that are “core outcome sets (COS)” that define a minimum of standardized outcomes every study should provide (COMET [23]). Their list of 68 outcome domains needs further compression to enable better comparison of future studies and is an ongoing project as stated by the authors [22].

According to Roaf [24], the prevention of PUs is a problem of “economic feasibility rather than a lack of preventive knowledge”. The high economic costs stem from the extensive effort required for manually monitoring the risk of PUs and taking preventive measures by hand. Technical assistance might ease the manual work and lower the cost. The majority of technical systems emphasize ways to prevent damaging skin pressure, such as using soft bedding or encourage frequent patient movement. Sheets or mattresses to measure the patient’s position, movement (e.g., [25,26,27]) or pressure (e.g., [28,29]) are often used to derive PU risk estimations. Another team of researchers tried to monitor patient positions in the bed with the help of wearable beacons, i.e., small wireless transceivers, and multiple sensors distributed in the room [30]. Special beds, mattresses or room installations share the disadvantage of requiring installation. This limits their use in ambulatory or wheelchair settings.

The advantage of wearable prophylaxis systems is their potential usage in the ambulatory setting and the ability to individually monitor patients during the whole day. Cicceri et al. [31] present a mobile, low-cost PU prevention tool based on motion measurements. A similar approach is used by Monroy et al. [32] who take advantage of motion data from an inertial sensor. However, those systems only try to estimate the risk of PUs by movement analysis. This is only a substitute measurement for the primary cause of PUs, which is long-term pressure or shear forces on the skin as described in Section 1.2. Silva et al. [33] reach a similar conclusion in their recent review on PU prevention methods: most intelligent and sensor-based approaches to prevent PUs are focusing on the classification of patients’ lying positions and may even lack real clinical studies. Furthermore, only very few approaches in the scope of their review actually integrate sensor-based data.

The authors could not find any preventive tools that focus on the actual PU development. As presented in Section 1.2, the most likely reason for the development of a PU is reduced tissue perfusion or skin that is pressurized over a long period of time. Hence, the most promising solution in our opinion is to monitor the tissue blood oxygenation during pressurization, to effectively prevent PU development. One comparable approach is used in Panahi et al.’s [34] study, which monitors SpO${}_{2}$ levels at diabetic foot ulcers, but focuses on wound healing and not prediction.

### 1.5. Tissue Blood Oxygenation—Measurement Methods

The pressure ulcer risk factor discussed above, tissue perfusion, correlates with tissue blood oxygenation (SpO${}_{2}$). Hence, we want to quantify the actual tissue SpO${}_{2}$ and use it to predict the occurrence of PUs. SpO${}_{2}$ measurements are not commonly applied for different types of tissue apart from the wrist or fingers, even though sensors theoretically permit monitoring different body sites. These measurements are often referred to as skeletal muscle oxygen saturation (StO${}_{2}$) and have different levels of precision, depending on the measured depth [35]. Other studies have used this technology to study tissue oxygenation in skeletal muscle, monitor compartment syndrome, or assess patients with lower-extremity arterial disease [36]. On the other hand, measuring blood-oxygenation with SpO${}_{2}$-trackers is well established as the noninvasive standard in modern healthcare and lifestyle such as smartwatches. Many studies have shown the accuracy of these devices, leading to their widespread use. One such study is provided by Lauterbach et al. [37] or the recent experiment by Kang et al. [38], showing a clear and precise correlation between inspired air-oxygenation and blood-oxygenation. Due to the widespread availability and adaptability of these SpO${}_{2}$ sensors, we considered them for the new application of local tissue-oxygenation monitoring. Having a temperature monitoring device directly adjacent to the SpO${}_{2}$ monitor is an integral part of our systems design. Studies have shown the strong correlation between tissue temperature and oxygen saturation in extremities, with lower tissue temperature leading to lower blood-oxygenation in that area [39].

### 1.6. Research Hypothesis

The underlying hypothesis for this research is that the development of a PU due to prolonged high skin pressure is indicated by insufficient tissue perfusion and can be detected before visible skin damage occurs by measuring the skin’s vital signs: local tissue oxygen saturation (SpO${}_{2}$) and skin temperature. To test the research hypothesis, we developed a wearable sensor device that continuously monitors these parameters at a predilection site. Figure 2 displays the proposed system architecture. Its development, verification, results from a clinical study and the evaluation of an early warning pattern are described hereafter.

## 2. Materials and Methods

This study investigates the development of pressure ulcers (PUs) with the goal to predict their early occurrence. Therefore, the following section introduces the developed electronics for data acquisition as well as the study population for the human trials (see ethical approval at the end). The acquired data presents an entirely new set of vital sign trends combined with pressure load measurements taken directly on the skin and is published on request [41]. The results generated in this study can be reproduced with the publicly available software [42].

### 2.1. Sensor System and Electronics

Since we want to record vital signs and pressure measurements in local proximity to the PU predilection sites of a human body, we had to develop a new sensor system. Usually, vital sign trends of interest are only recorded with a finger pulse oximeter, in a smartwatch or an ear thermometer (SpO${}_{2}$, HR, temperature), but not at body tissue elsewhere. The authors are not aware of any standard measurement for pressure load at predilection sites related to PUs.

Figure 3 displays the KIPRODE sensor system: as shown in Figure 3a, it consists of the main electronic (1), the pressure sensor foil (2) and sensor node (3). The latter is a button-shaped device measuring SpO${}_{2}$ in the adjacent tissue, heart rate (HR), and skin or local tissue temperature, respectively. The pressure sensor foil is a thin and flexible foil, with a thickness of less than a millimeter, measuring 6.5 cm on each side and equipped with 25 force-sensitive elements. Those are conductive, printed elements with a linear resistance changebased on the applied force that can be measured analogously. Technically, it is a measurement of force per area. However, due to the consistent element size and for the sake of simplicity in terminology (relating to pressure ulcers and pressure measurement), we continue to use the phrase. Combined, both devices can fit in the palm of a hand. These measuring devices are connected to a shared box by separate cords, that are between 25 cm and 35 cm in length. The box measures approximately 10 cm · 6.5 cm and contains the hardware required to run the system, such as the microcontroller and sensor control as well as a rechargeable battery and SD card to save measured data. The small size allows for continuous, highly targeted measurements without imposing a significant burden on the patient or the medical staff.

The main electronic consists of a microcontroller that calls every sensor (pulse oximeter, skin temperature and pressure) in a periodic interval (5 min for the pressure sensor and 1 min for all others) and saves the data onto an SD card. In a next development stage, the data should be sent via Bluetooth to a connected device. For ease of engineering and due to the absence of compatible smartphones during the trials, we opted to use SD storage.

For manufacturing, first the circuit boards are assembled and tested. That comprises the main board (1) with most electronic components and the sensor node (3) holding the skin temperature sensor and pulse oximeter. The pressure sensor foil (2) has a pin header to easily connect it with the electronics. All wire connections are implemented with adjusted cords and protected against impacts of touch as well as humidity with heat shrink tubing of different sizes. To enable the exchange of broken wires or damaged sensors, the wires are plugged into the main electronics that reside in a plastic housing (white in Figure 3a). The gray colored plastic element implements our strain-relief system to protect the plugs from external strain when a patient pulls the cords. The cords are wrapped around the plastic element such that it takes external strain. A white plastic cover is then screwed onto the main housing in (1) to complete the electronic system.

The pulse oximeter is the MAX30102 chip from Analog Devices [43] and represents a common state-of-art device that is used in many smartwatches. Ahmad et al. [44] report a high accuracy of this pulse oximeter module comparing it with hospital patient monitoring: SpO${}_{2}$ measurements align with 97.6%. A pulse oximeter consists of at least two LEDs, an infrared and a red one. These LEDs send light pulses onto and into the skin, which are partially reflected. Oxygen-saturated blood molecules show a different reflection pattern than those not carrying oxygen. By analyzing the reflection pattern of both LEDs, one can determine the oxygen saturation of the blood. In well-known finger-tip devices, the light passes through the whole finger and the pattern of light that passes through the finger is analyzed. This method is limited to body sites where only a short distance through the skin has to be overcome such as fingers or toes. A reflexive pulse oximeter as used here, on the other hand, measures the light reflection and thus can be placed anywhere on the skin. Depending on the type of local blood circulation, one can read the tissue oxygenation at several locations of the body. Since we want to investigate the blood circulation at the hip (trochanter) and back (sacral), this method was selected. The authors are not aware of a similar approach to determine tissue oxygenation at the trochanter and hence develop this new method and validate its performance in Section 3.1. Skin temperature measurements are performed by a MLX90632 sensor from Melexis [45]. It obtains temperature readings via infrared measurements eliminating the need for direct skin contact. According to the manufacturer, it gives an accuracy of ±0.2 °C in the human body temperature range.

Data streams from our KIPRODE system for around six hours are presented in Figure 4. Figure 4a shows all relevant sensor values continuously, whereas Figure 4b displays the five by five sensor elements of the pressure sensor foil for different moments that are marked with red lines above. Especially, SpO${}_{2}$ measurements can be easily disturbed and lead to measurement noise, for example because of patient movement or distracted scattering due to a changed sensor distance to the skin; a moving average filter is used to mitigate noise (more details in Section 2.5.1).

Additionally, we calculate a validity factor for each reading that analyzes the raw values of each light sensor and only gives a validity statement if the values appear reasonable. The positive outcome is that we can read tissue blood oxygenation at predilection sites (here: the trochanter) with acceptable quality and reliability as shown in Section 3.1.

### 2.2. Data Pipeline

Within the project, we collected time continuous sensor data as well as corresponding medical data from the physicians. The medical data contains physical data, relevant preexisting conditions, vital sign data recorded by the medical staff as well as dates and results of the Phillips finger tests (see Section 1.3).

All recorded data needs safe and privacy-ensuring transfer and storage. Therefore, we utilize the so-called “Pflegekonsil” which is an online database for medical data in the German healthcare system and is provided by our partner Monks [3]. It holds several privacy proofing certificates. Their online system provides various levels of access, ensuring that only registered physicians can identify patients and see their personal data. During the study, the medical study personnel reads the acquired sensor data from SD cards and uploads it into the “Pflegekonsil”. There, Fraunhofer researchers only have access to fully anonymized data, comprised of sensor data streams from the KIPRODE sensor system as well as the medical diagnosis and physical data. In future developments, this should happen automatically via a connected device such as a smartphone.

### 2.3. Medical Data Acquisiton

The study was carried out by the clinic and polyclinic for plastic surgery and hand surgery at the Rechts der Isar clinic in Munich (MRI). The study was designed to primarily include bedridden patients that are likely to stay in the stationary clinical setting for up to two weeks. Because these patients are bedridden, they have an increased risk of developing pressure ulcers during their stay, compared to mobile patients. Eligible patients are selected and invited to participate by medical doctors working in the plastic surgery team at the clinic. Participating patients got a profile including physical data and preexisting conditions. During the study, we documented their daily vitals obtained with gold-standard measurements (blood pressure, HR, blood oxygen saturation at the finger tip, and body temperature in the ear) together with corresponding timestamps.

To identify the best position for the sensor placement, every patient was individually examined and asked about their preferred lying position. Based on those results we attached the two sensing devices (vital signs and pressure sensors) on the predilection site experiencing the highest pressure, meaning that the patient was lying on that site for a large majority of the time. The chosen sites were most commonly the sacrum or spina iliaca superior posterior when the patient’s main resting position was on their back. If patients had to lie on their side, we typically placed the sensory devices on the trochanter major facing the bed. The sensory devices were attached with a Mepilex plaster with a gap of 1–2 cm between each device. We paid special attention to to ensure consistent sensor placement for each patient.

To maximize the battery run-time and still have quasi-continuous measurements by the sensor, we had to compromise on the measurement intervals. Given that pressure measurements require more energy than the other sensors, we used a five-minute sampling frequency, while the other sensors were read every minute. This was not considered problematic because pressure changes are less frequent than HR or SpO${}_{2}$ changes in bedridden patients. For medical classification of the skin health, trained medical personnel checked the predilection site under the plaster twice every day. The Phillips finger test was conducted as mentioned above (see Section 1.3) and the result was documented with timestamps. Given that a positive Phillips finger test characterizes a stage I PU, we can then evaluate the data since the last negative test, leading to a better understanding of the exact pathology of the illness. Patients with a positive Phillips finger test were immediately excluded from the study in order to properly treat the newly developed pressure ulcer.

### 2.4. Study Population

The observation of patients in this study was time consuming and required commitment from the patients. Due to discomfort with the sensor system, a number of patients withdrew from the study. We tried to use as much of their collected data as possible. Every long-term patient at the MRI was considered a potential participant if their health status was stable and they where able to contribute. Most of them had to stay for up to two weeks and were in care because of a wound or operation, hence they have had preexisting conditions already. Table 1 shows the numbers of participating patients and how long they contributed to the study. We can see a decrease in numbers from patients that had started and those who completed the full 10-day duration. While some patients recovered faster than expected, others expressed discomfort with the sensor system. A revision of the latter should enhance the comfort of the patients.

An ideal control group would exhibit physical data very similar to that of the patients group only without the risk-influencing preexisting conditions. However, creating such a group would involve elderly individuals who might not be accustomed to electronic devices and could be challenging to recruit outside the hospital. Therefore, we opted to work with a healthier group of individuals who responded to our notice in the hospital. This notice was positioned in the MRI asking for people to join the control group study. Nevertheless, the control measurements allow for the verification of the system functionality (Section 3.1) as well as comparisons between healthy subjects and patients with preexisting conditions (Section 3.2).

The study population can be described by their physical data such as age, height and weight as well as preexisting conditions. Figure 5 gives an overview of the physical data and compares patients with the control group. As discussed above, there are some difference between the control and patient group (see Figure 5a). On average, the control group is significantly younger and exhibits a lower body mass index, indicating a more sportive condition. This factor could potentially influence the comparability of the data and is taken into account during analysis and discussion.

Pulse oximeters measure optically which means that light scattering can influence the recorded results. This can also occur with higher pigmented skin types and potentially distort SpO${}_{2}$ records as reported in literature [46,47]. The study was performed in a hospital in Munich, where the majority of patients have lower skin color types according to Fitzpatrick scales: most are within Fitzpatrick types 2 and 3.

A more detailed analysis of preexisting conditions within the patient group is presented in Figure 5b. The graphics show the overall study group with both, control and patient group. On the far left, one can see that approximately one-fourth of all participants had already been diagnosed with diabetes, which is assumed to impact the PU risk. Another strong risk factor is that almost half of the study group is categorized as immobile (indicated by the green color); the healthy control group is not immobile (light blue). Immobile patients are bedridden and only possess limited mobility. Therefore, they rely on healthcare workers to move them in order to better distribute pressure and prevent PU development. Those individuals are at higher risk for PU development because turning happens on a periodic schedule based on experience and not on their actual necessity. This is an area we aim to address with a PU prediction device. Paraplegic patients are even more restricted in their movements and hence need more extensive care. While it might be feasible to provide such a level of care in a hospital setting, it is difficult to replicate in patients’ home environments, which is often leading to repeated hospitalization after release. A pressure ulcer with or without a surgery prior to the study enhances the risk for a new PU generation; thus we show their frequency in the two graphs on the right in Figure 5b.

### 2.5. Data Analysis

The following section introduces the data processing steps used to format the data for machine learning methods. Further insight into these methods can be gained by reviewing the publicly available software [42].

#### 2.5.1. Preprocessing

First, all data files are uniformly formatted. Then the data of every tested person is aggregated, missing timestamps are filled up with blank values, and out-of-range sensor data is deleted. Missing sensor values of the skin temperature, the heart rate, and the SpO${}_{2}$ are forward-filled to a maximum of five minutes. The amount of valid values in the preprocessed dataset can be seen in Figure 6a. Since the HR and SpO${}_{2}$ are calculated from the same raw sensor signals, their proportion of valid values is similar. The used skin temperature sensor is more reliable, thus almost all temperature values in the preprocessed dataset are valid.

Next, the data is split into relevant sections. This is performed by introducing a data quality factor that calculates the fraction of valid sensor data per timestamp. Sections of data with a data quality factor higher than a preset threshold are marked as relevant. The distribution of the lengths of relevant sections can be seen in Figure 6b. Lastly, the relevant sections of every patient are saved in a sktime format data frame [48]. The mean length of patients’ sections is 990 min, but many datasets are considerably shorter, leading to a median of 395 min.

#### 2.5.2. Relationship between Skin Temperature and Pressure Loading

The high data quality of the skin temperature measurements allows us to further investigate how the skin temperature changes in response to varying pressure loads of different duration. Therefore, we adopt the following approach: first, we locate notable events in the pressure time series using peak detection algorithms. Subsequently, we group these events based on their duration using clustering techniques. To examine the skin temperature reaction to pressure events of different duration, we employ spike-triggered averaging (STA). This process allows to gain insights into how skin temperature responds to variations in pressure over varying duration periods; a comprehensive description follows hereafter.

First, the time series data are smoothed using a Butterworth filter to enhance its resemblance to a plausible physiological process and to eliminate noise. Clustering is sensitive to data scales, therefore we standardize the data through z-score normalization and transform it to have zero mean and unit variance.

After that, we used the popular scipy library [49] to identify pressure peaks. The library defines a peak as any time-point whose two direct neighbors have a smaller amplitude. If multiple consecutive points have the same amplitude, the one in the middle is returned. The detected peaks are then filtered by peak distance (minimal distance between two peaks), peak prominence (relative height of the peak) and peak width (width at half of the peak’s prominence). Around the detected pressure events, we extracted 60-min windows, starting 5 min before and ending 55 min after the pressure event.

To distinguish the temperature reaction to pressure load events of different duration, we used machine learning-based clustering. There are a multitude of different clustering techniques for different purposes. We evaluated several time series clustering algorithms on an annotated subset of our detected pressure events. The best overall performance was achieved using time series k-means with input scaling and with Euclidean distance as distance metric. Time series k-means is based on the popular k-means algorithm [50]. It is simple, efficient, performs well, and is used across a wide range of applications. After randomly initializing K cluster centers, it iteratively assigns each sample to its closest cluster by minimizing the within-cluster sum-of-squared distances. To adapt the k-means algorithm for time series, the series are flattened into a table, treating each time index as a separate feature. For our analysis, we used the implementation provided by the tslean library [51]. To determine a number of clusters K that represents the intrinsic structure of the underlying data, we used the so-called elbow method [52].

To analyze the impact of the detected pressure events on the skin temperature, we needed a technique that summarizes the temperature measurements after the pressure events over all the segments in each cluster. More precisely, we wanted to compare how the skin temperature changes after a pressure event relative to a reference point right before the pressure event. A suitable approach is spike-triggered averaging (STA) [53]. Our modified version of the algorithm is described in more detail in Section A.2.

## 3. Results

First, we verify the sensor validity by comparing the measured values with the clinical reference and common sense in Section 3.1. Section 3.2 shows the influence of preexisting conditions on vital signs within the study group. Finally, we introduce our most important findings that suggest the possibility to build a PU predictor in Section 3.4. Moreover, a parallel study on PUs at feet is presented in Section 3.5 as it only requires little system adaptation.

### 3.1. Verification of Kiprode Sensor System

To verify the KIPRODE sensor system and ensure proper sensor operation, all recorded data files are compared to gold standard measurements of today’s medicine whenever available. Basically, two different ways of sensor verification are used: first is the comparison with medical reference data that is recorded twice a day during the study by medical personnel (medical reference). The second way is to compare the measured data of patients with the control group and set it into relation with expected values (common sense).

#### 3.1.1. Kiprode Sensor Data vs. Medical Reference

The primary values that we measure and analyze within this research are SpO${}_{2}$, heart rate, skin temperature and the pressure load. There is no medical reference for the latter and its functionality is proven with adjusted calibration in the lab.

A practical way to assess the functionality of the sensor system involves comparing it with the medical screening data that the personnel documented daily. Not all values can be directly compared as no standard reference measurement exists. We evaluate the sensor system by comparing the distributions of sensor data and reference data. The biggest challenge is that we measure at patients’ hips or back (trochanter or sacral) but standard reference measurements are performed at finger tips (heart rate and SpO${}_{2}$)) or in the ear (body temperature). A comprehensive comparison of the above described values is provided below, based on Figure 7. The dataset of KIPRODE sensor data is far larger for all data types, as the system measures continuously, but medical reference required manual interaction and hence is only determined on specific times.

There are plausible screening data for heart rate measurements that allow for direct comparison with the KIPRODE sensor data as the heart rate is the same, independent of the measuring position. According to Figure 7a, both graphs show a similar distribution with a median that only deviates by one pulse per minute. That slight difference might occur because reference data is taken at specific times, whereas the KIPRODE system measures continuously. The comparison provides a good indicator that the KIPRODE heart rate measurements are working reliably.

On the other side, the reference temperature in the medical dataset represents the body temperature that is taken at the patients’ ear whereas the KIPRODE skin temperature is derived close to the pressure sensor at the patients’ hips. Consequently, the temperature readings cannot be compared with absolute values and we rather look at their distribution in Figure 7b. Nevertheless, the temperature readings seem to be similarly distributed for both measurement methods, matching typical pathophysiologic patterns.

However, for SpO${}_{2}$ measurements, the gold standard is to determine the value at finger tips or toes. As we measure at the hips or back of a patient, no direct comparison is possible. The second type of sensor validation is based on the comparison with the control group as described below. Figure 7c displays the recorded data with a notable difference between reference and KIPRODE data. If we take into account the different measurement sites at patients’ bodies, we obtain very interesting results: the SpO${}_{2}$ values for patients within this study seem to be notably reduced (mean of 91.7%) at the hip compared to 95.6% at their fingers. That is reasonable from a medical perspective as those patients may have worse tissue blood circulation due to their preexisting conditions. We are confident to trust the KIPRODE sensor readings because of the comparison with the healthy control group in Section 3.1.2. Hence, the study already shows an interesting finding: people with higher risk for developing a PU show strongly reduced SpO${}_{2}$ values at their hip tissue compared to their finger tips.

#### 3.1.2. Vital Sign Analysis of Patients and Control

Additional insights into the quality of sensor data can be obtained by comparing the sensor data from patients with the control group as well as with medically expected values.

Figure 8 shows the mean and standard deviation of all recorded vital sign values (SpO${}_{2}$, heart rate, skin temperature) for the control group compared with patients. The vital sign analysis of the control group meets the expected ranges and suggests a proper operation of the KIPRODE sensor system. SpO${}_{2}$ values should be close to 100% for a healthy person and the mean heart rate of around 60 beats per minute is expected as well. There is little comparison for skin temperature readings at the hip, but generally speaking values between 35.0 °C and 36.5 °C are in a normal range. Hence, all three vital sign distributions of the control group lie in the expected range, according to the medical professionals of the MRI. The orange dots mark both patients that developed a stage one PU during the study. Further analysis of their skin health follows in Section 3.4.1. Their average vital signs show no extremely conspicuous values compared to the whole patient group.

Setting the patient group into relation with the control provides an expected result for the heart rate in Figure 8a. On average the patient group is older, less sportive and suffers from more preexisting conditions. All factors correlate with an increased HR and hence the HR values are not surprising ($70.2\pm 10.8\;\frac{beats}{min}$ for patients versus $57.4\pm 7.0\;\frac{beats}{min}$ in the control).

Figure 8b displays a higher skin temperature for patients (36.0±0.7 °C) compared to the control group (35.3±0.4 °C). Since the absolute skin temperature is influenced by external parameters such as blankets during night or room temperature, we do not evaluate the absolute value and interpret relative skin temperature changes per individual only (see Section 2.5.2).

Of particular interest is the significantly reduced mean SpO${}_{2}$ within the patient group in Figure 8c. The mean SpO${}_{2}$ value is significantly reduced at 91.3±2.9% compared to the mean of healthy individuals, which is 98.5±0.9% here. The latter value is considered normal for healthy individuals, which validates the accuracy of the KIPRODE sensor data. The mean value of around 91.3±2.9% is also reduced compared to the finger tip measurements in Figure 7b, which reveal a mean SpO${}_{2}$ of 95.6%. Those findings are a reasonable indicator that the mean blood oxygenation in the body tissue of high PU-risk patients differs significantly between the extremities and the central body, which is not the case for healthy subjects. A possible explanation is that damaged tissue suffers from the pressure load and leads to worse tissue perfusion. This supports our research hypothesis and opens the possibility to predict PUs in early stages, preventing further damage.

We investigated the mean pressure load that subjects were exerting on their predilection sites (trochanter and sacral): Figure 8d gives no explanation for the above described differences in body vital signs as the distribution of mean pressure load is very similar in control and patient group.

### 3.2. Effect of Preexisting Conditions on Vital Signs

This research investigates the correlation between tissue blood oxygenation, skin health and PU development risk. Here, we looked at the influence of preexisting conditions on the body vital signs in the patient group. Preexisting conditions that are commonly associated with a high PU risk are, among others, paraplegia, immobility, diabetes, or if a patient already had a PU or even a PU-related surgery. All of them affect the body tissue SpO${}_{2}$ similarly and are often correlated with each other. Hence, we focus on the interpretation of the effects a previous PU surgery has on body vital signs in Figure 9 and show the others in Section A.3.

Figure 9a does not indicate a significant influence of preexisting conditions on the heart rate compared with similar patients that did not undergo a PU-related surgery; stronger correlations are observed with physical conditions (see Section 3.1.2). However, both skin temperature as well as body tissue SpO${}_{2}$ suggest a negative correlation with a previous PU-related surgery (Figure 9b,c): both are significantly decreased for patients with a previous surgery. The mean skin temperature decreases from 36.2 °C to 35.5 °C and the SpO${}_{2}$ decreases from 92% to 89.8%. Those findings align well with our research hypothesis assuming reduced tissue blood perfusion for patients with preexisting conditions. A lower average skin temperature could be a reasonable consequence of disturbed tissue blood flow. Comparing the skin temperature between patients is valid as they experienced very similar conditions within the hospital, even if we cannot compare it with the control group above. The same conclusion about damaged tissue perfusion can be drawn from reduced tissue SpO${}_{2}$ averages as they are the primary parameter allowing to quantify oxygen transportation into the tissue. Again, we compared the pressure load in both groups (with or without previous PU surgery) without observing any relevant difference. This means we see significantly deteriorated tissue perfusion for patients who already had a PU.

### 3.3. Relationship between Skin Temperature and Pressure Loading

The skin temperature is affected by underlying tissue perfusion as high blood circulation leads to increased skin temperature. The methods to analyze the relationship between skin temperature and pressure loading were elaborated in more detail in the master’s thesis of Schillinger [54].

The assumption that changes in skin temperature are linked to the risk of pressure ulcer development has been demonstrated in prior studies [55,56]. In most cases, those measurements were taken at distinct times because continuous monitoring was too challenging at relevant body sites. However, with our KIPRODE sensor system, we can observe the skin temperature of at-risk patients during the whole study continuously (with measurements every minute). This enables us to gain further insights about how pressure load and skin temperature relate to each other.

The approach within the study is to investigate skin temperature values directly after a notable pressure change is detected. The latter would probably occur due to a patient changing their position from one body site to another; hence relieving the previous body site from pressure load. By utilizing machine learning methods (as described in Section 2.5.2), we identify those relevant pressure changes within all recorded pressure and temperature data.

Figure 10 shows the analysis of temperature variations in relation to changes in pressure. Sudden increases in pressure load are presented in Figure 10a and are likely associated with a patient shifting their weight onto the sensor, whereas Figure 10b displays the pressure load relief. The left graphs of both figures represent the patient-normalized pressure load with a dotted line showing the moment of the detected pressure change. The corresponding graphs on the right show relative temperature trends after the detected pressure change. Detected pressure load changes are categorized according to their length: 20, 40 or more than 55 min of lasting pressure load. The available data was insufficient to effectively investigate longer time periods.

The majority of the graphs show a strong positive correlation of temperature trend and pressure load. Thus, when weight is shifted onto the sensor, i.e., the predilection site, the skin temperature increases with minimal delay. This is likely due to the compression of the skin, which rapidly increases tissue perfusion, subsequently raising the skin surface temperature. We can see the opposite effect for pressure relief in Figure 10b.

For the first time, we show the continuous skin temperature in conjunction with actual skin pressure loads at PU risk patients. Those are short-term reactions of the tissue. Further optimization to detect deteriorating tissue perfusion over longer time spans could enable PU risk estimations based on skin temperature as proposed in other studies [55,56].

### 3.4. Development of a Pressure Ulcers Predictor

The goal of this study is to develop a pressure ulcer predictor that will alert patients to prevent PUs and their severe consequences. Since we tried to avoid increasing the risk of PU development among patients, we followed the clinical practice for PU prevention during the study. As a result, we only had two positive diagnoses from the Philips finger tests, indicating a stage one PU (see Section 1.3). In one case, useful sensor data is lacking due to technical or application issues. Hence, we focus on the analysis of the well-documented case. In order to classify and detect the actual development of a pressure ulcer with statistical relevance, a higher number of positive samples would be required. Therefore, training of supervised machine learning algorithms is not possible. Nevertheless, the analysis presented in Section 3.4.1 yields promising results. The proposed predictor is tested using the acquired study data in Section 3.4.2.

#### 3.4.1. Analysis of Diagnosed Pressure Ulcers Development

As discussed in Section 1 disrupted tissue blood oxygenation is assumed to lead to deteriorating skin health and this might be detected before skin damage is visible by a decrease in measured SpO${}_{2}$ values.

Figure 11 displays the recorded SpO${}_{2}$ values at the predilection site within the six hours before a stage one PU was diagnosed by medical staff. Starting around 3.5 h prior to the PU diagnoses, the 1 h rolling mean of SpO${}_{2}$ steadily falls and decreases from around 94% to 87% within 1.5 h. After that, the signal quality is too low to calculate a representative rolling mean. This decrease in mean SpO${}_{2}$ is consistent with our research hypothesis. As the the diagnosis of a stage one PU occurs two hours after the decrease in mean SpO${}_{2}$, we could raise a warning to prevent the development of a severe PU.

If this indicator can be validated in a larger study group it could potentially lead to the development of a wearable PU prophylaxis system. However, this requires further research.

#### 3.4.2. KIPRODE Pressure Ulcers Predictor

Based on the results in Section 3.4.1, we propose an early warning algorithm that could warn patients before an actual PU develops. Since the study group data only contains sensor data of one PU development case, we are constrained to evaluate our proposed predictor on the non-positive cases. While this does not validate our predictor, it does show how often this pattern is present in the study group data. To reliably verify this predictor, a larger number of positive cases is required. Therefore, a study group with a higher risk of PU development is needed. To encourage further research, we share the code that has been developed so far [42].

We assume the rolling mean of SpO${}_{2}$ to be a strong PU prediction parameter, because it represents the trend of SpO${}_{2}$ measurements. Therefore, a warning is raised, if the 1 h rolling mean of SpO${}_{2}$ is significantly lower than the expanding mean of SpO${}_{2}$ of the patient data available at that time:(1)$${\mathrm{diff}}_{{\mathrm{SpO}}_{2}}={\mathrm{mean}}_{60\min}\left({\mathrm{SpO}}_{2}\right)-{\mathrm{mean}}_{\mathrm{patient}}\left({\mathrm{SpO}}_{2}\right)$$
(2)$$state\left(t\right)=\left\{\begin{array}{cc} warn, & \mathrm{if}\;{\mathrm{diff}}_{{\mathrm{SpO}}_{2}}\leq \mathrm{thres}\\ normal, & \mathrm{otherwise} \end{array}\right.$$
the rolling mean over 1 h is only calculated, if at least 90% of the SpO${}_{2}$ values in the window are valid. The resulting distribution of differences of the means can be seen in Figure 12b. The warning threshold is set to −3%. These parameters are chosen based on medical estimations and heuristic data analysis. When a dataset with more positive cases is available these parameters need to be optimized with respect to the receiver operator characteristic.

A more detailed look into the recorded data of the patient 10 h before the stage one PU was diagnosed is given in Figure 12a. Here, one can see the skin’s vital signs as well as the sum of the pressure applied on the skin. Because we are measuring continuously on moving patients in a hospital environment, the sensor data is noisy. For visualizing the trends of the data, we apply filters as described in Section A.1. The positive Phillips finger test was recorded at the red dotted line on the right. Therefore, we are searching for a pattern that occurred prior to the positive result and serves as an indicator for a developing PU.

Our proposed predictor searches for a significant decline in the 1 h rolling mean of SpO${}_{2}$ below the expanding mean of a patient (see Equation (Equation 1)). In the displayed data, a warning would be raised by the decline in SpO${}_{2}$ inside the red area. This would enable the patient or a caretaker to prevent a serious PU. Figure 12c displays the number of positive alarms that our predictor would have generated within the study group. The patient, who was diagnosed with a PU, would have received six alarms (green). We cannot confirm the accuracy of these predictions because we do not receive the medical diagnosis continuously, but only when a doctor examines the patient. In a follow-up study, the medical staff should check for PU formation every time a warning was raised by the algorithm. This way, the accuracy of our proposed predictor can be evaluated. The fact that also some control group members would have received a warning indicates that some false alarms are raised. As discussed in Section 4, pressurizing healthy skin would not lead to low tissue perfusion. Analyzing the raw data of alarm events within the control group suggests faulty measurements. The control group applied their sensor systems themselves, while the patients had them applied by medical staff. Loose sensor attachment and more individual movement can lead to motion artifacts, which are stronger for healthy people. These motion artifacts can distort pulse oximeter readings, resulting in incorrect measurements. To filter out the motion artifacts in future designs it could help to add an accelerometer.

### 3.5. Pressure Ulcers at Patients’ Feet

Pressure ulcers can occur on different sites and in various types of patients. The first part of the study addressed PUs in hospitalized, bedridden people at high risk of PUs at their hip or lower back. Similarly, diabetic patients have an increased risk for PU development at their feet; referred to as diabetic foot ulcer (DFU).

As described in Section 1.2, a PU can arise from an intense pressure load on preconditioned skin and lead to skin breakdown and further injury including infection or even amputation [57]. Due to neurological insensitivity at the peripheral level, individuals with advanced diabetes are particularly susceptible to pressure ulcers as their perception of pressure and pain is weakened in the extremities [58]. The most common predilection sites for these individuals are the feet, particularly the heels, ankles, and toes. Unfortunately, foot ulcers in diabetics are highly susceptible to infection, resulting in diabetic foot infections (DIFs). Diabetic foot infections are associated with increased morbidity, daily wound care, high healthcare costs, and surgical intervention. A 12-month prospective observational study by Ndosi M, Wright-Hughes A, Brown S, et al. [59] provides a prognosis for patients with infected diabetic foot ulcers (IDFU). Their study showed that after one year, only 46% of affected patients had their foot ulcers healed, out of which 10% recurred. Fifteen (15)% of patients died during this time and 17% required lower extremity amputation, reflecting the severity of this condition [59]. About 8–15% of patients with diabetes will experience feet PU once in their lifetime. In Germany, every 13.4 min a diabetes-related amputation is performed [60].

In 2016, about 18.6 million people worldwide were affected by DFUs alone and 6.8 million had to live with an amputation of lower extremities [61]. This corresponds to a worldwide prevalence for DFU in 2016 of 270 per 100,000 people and 96 for DFU-related amputations, respectively [61]. Armstrong et al. [62] provide a detailed cost analysis for DFU treatment: the cost of diabetes increased by 26% in the US from 2012 to 2017. Of the US$237bn spent on diabetes in 2017, it was estimated that a third was needed to treat DFU. This puts the economic cost of DFU treatments at the same level as the direct costs incurred in US cancer treatments in 2015; those amounted to approximately US$80.2bn. They also show a almost similar 5-year mortality for patients with a DFU as for cancer patients (30.5% [62]).

In the KIPRODE study, we modified our system to also investigate those PUs, considering their medical proximity. In this study, we did not monitor the development of any PU. Hence, the investigation focuses on the prophylaxis of the main risk factor for DFUs: pressure peaks at patients’ feet. It is widely assumed that individually adjusted orthopedic shoes offer the most effective protection against these risk factors. We investigate this hypothesis by continuous feet pressure load monitoring.

#### 3.5.1. System Adjustment for Prophylaxis of Feet Pressure Ulcers

To prevent the severe consequences of feet ulcers, we have to adjust our KIPRODE system as described in Section 2.1: the pressure sensor area is transformed into a feet-like shape (4 by 12 rows of pressure sensitive elements) which can be placed inside a shoe as depicted in Figure 13. The sensor node measuring vital signs has to be placed at the lower leg to be as close as possible to the foot. It cannot be inserted into the shoe to not introduce any additional PU risk because of rims or edges. In this stage of development the sensors are connected via cords with the main board (see Section 2.1) which is not the optimal solution as it can be inconvenient for the patient. Future optimizations should consider the implementation of wireless transmission such as Bluetooth or NFC for data transfer. A difference to the previous study design is that we are observing patients while walking or standing, rather than while lying down. That means, pressure loads are usually not of similar duration and PU development is more likely to occur due to repetitive pressure peaks at the feet.

Neuropathy leads to a reduced or diminished sense for pressure load at patients’ feet. That can lead to unnoticed developments of pressure ulcers at their feet. Hence, a reasonable approach to reduce the risk for foot ulcers is the reduction in pressure peaks. This has already been tried through the use of orthopedic shoes that are precisely fitted to the patient.

The main research question is whether an individually adjusted orthopedic shoe can reduce the risk for the generation of foot ulcers. As the study does not expect the actual formation of foot ulcers, the parameter under investigation is the pressure load on the feet during the daily activities of ambulatory patients. The goal is to determine abnormal pressure load patterns by machine learning algorithms and warn a patient if such is detected.

#### 3.5.2. PU-Risk Mitigation by Individual Shoes

During the study conducted at Fußnetz Bayern (FNB) [63], half of the patients wore orthopedic shoes, while the other half wore regular shoes. This was then compared with a healthy control group.

The whole patient group comprises ambulant patients that are already regularly visiting the doctor because of a previous ulcer. In this study, the KIPRODE sensor system is applied to the opposite foot, where no ulcer has occurred previously. Those patients are at a high risk of developing further ulcers and are therefore motivated to mitigate the risk as far as possible. For the adapted study, we included 13 ambulant patients for a period of more than two weeks and nine healthy control subjects for a period of more than one week. The duration of the patient study varied strongly (ranging from a few days to more than three weeks), depending on patients’ commitment and comfort.

To evaluate the pressure load on patients’ feet, we measure the pressure load inside a shoe continuously over the observation period. Therefore, every minute a pressure scan is performed across the whole sole, including 48 elements (4 by 12 rows) and saved with the vital signs at their legs. Since obtaining high-quality vital sign measurements proved to be challenging with ambulatory patients, meaningful interpretation is not feasible, and we focus exclusively on pressure load data. As we want to compare the pressure load between patients wearing orthopedic shoes and those without, we split the group accordingly, as shown in the graphs in Figure 14. In Figure 14a, it is evident that the pressure peaks as well as the mean pressure load are notably decreased for patients wearing an orthopedic shoe compared to those without or the control group. A more detailed analysis divides the sole of the foot into forefoot, midfoot, and rearfoot, as presented in Figure 14b–d. Figure 14d discloses the highest differences in pressure load mean and peak at the heel. The heel is also where the largest pressure loads usually occur while standing. Even though the mean pressure loads of patients with orthopedics in Figure 14c is slightly higher compared to those without, the peaks are reduced. The absolute pressure load on the midfoot is much lower than in rear and front. Hence, the impact of the midfoot is not as strong as the forefoot or heel, which aligns with other research such as Jasiewicz et al. [64]. At the forefoot, Figure 14b, pressure peaks are at their lowest for patients wearing orthopedic shoes, and there is a slight reduction in the mean pressure load as well. In summary, we reach a clear conclusion: orthopedic shoes can provide substantial support for the feet of patients with preexisting conditions by reducing pressure loads.

## 4. Discussion

In order to obtain the necessary data for the intended design of a PU prediction and prophylaxis system, we needed to develop a new sensor system. The authors are unaware of any comparable sensor system that allows the recording of body vital signs and pressure loads in humans. We focused on bedridden patients with an increased risk of PU development, hence, the system is optimized for placement at the hip or back (trochanter or sacral). Nevertheless, it could be adjusted for other body sites as well. The introduction of a new and highly integrated electronic device invariably brings along certain vulnerabilities and requires multiple rounds of improvement. For this reason, the sensor was constantly refined and optimized for clinical use. Although the method of measuring data remained consistent throughout the entire study to ensure data reproducibility, the hardware box was modified for ease of use. Examples of these optimizations include the implementation of double coverings and the extension of individual cables, along with the installation of strain relief at the connection points between the cables and the box. These changes have made the daily maintenance of the sensors much easier and increased their mechanical stability. Numerous code adjustments were required, which led to data loss at the beginning of the study. However, after some iterations, the KIPRODE sensor system was operating reliably and could be used for our purposes. The main findings include that patients with severe preexisting conditions require even simpler-to-use devices than the healthy control group. After a successful control group study, we encountered several system failures, for example due to removed cables, humidity or other mechanical stress. That resulted in some data loss until we increased the mechanical stability. For future developments, this should be taken into account from the beginning.

The long-term goal is to provide those prophylaxis tools to at-risk patients at home. As evident from the study, the system still requires a high level of monitoring and supervision by trained individuals. Some patients quit the study because the sensor system was not comfortable enough. This indicates that following the completion of functional development, a few additional comfort-related optimizations are required.

A major challenge when building a prophylaxis tool lies in the conflict between the necessity to monitor the disease and the simultaneous attempt to prevent its occurrence. In this case, it has resulted in only one well-documented and diagnosed PU in our study. Consequently, no statistically significant analysis is possible for our observations and we have to rely on qualitative analysis by comparing our findings with medically expected conditions. For future studies, we recommend to include hospital stations with an even higher risk for PU development, such as intensive care units.

The patient who developed a PU exhibited a noticeable decline in the mean SpO${}_{2}$ in the hours before it became visible. In our opinion, the reduced SpO${}_{2}$ values strongly correlates with the health condition of the tissue. That is also indicated by comparisons of tissue SpO${}_{2}$ values between patients and control group. Hence, we assume deteriorating tissue perfusion as a consequence of an ongoing pressure load due to the patient’s preexisting condition.

On average, the control group is younger and lighter than the patient group. Ideally, the only distinction between the control group and patients would be their preexisting conditions. However, recruiting healthy individuals with comparable physical characteristics to the patients is challenging. We publicly searched for volunteers on the MRI website and on the hospital’s white board and received the control group as presented above. The significance of certain findings is constrained by these differences between the two groups. However, since we validated our measurements as described in Section 3.1, it is unlikely that external factors or noise significantly distorted our conclusions. This means that physical effects cannot solely explain the great differences in vital sign measurements presented Section 3.1.2. Only the heart rate may be increased for patients due to their age and preexisting condition, but we do not focus on that. Skin temperature could be heavily influenced by environmental conditions, but our analysis solely examined relative temperature differences with similar conditions. Most importantly, the SpO${}_{2}$ data are validated with common-practice reference measurements in Figure 7; otherwise, if physical data would be the main influencing effect, the finger tips readings would have to show that, too. The variations in the decreased SpO${}_{2}$ readings at patients’ hip and back tissue (trochanter and sacral), most likely arise from their preexisting conditions that are also related to PU development. Further evidence is given by Gómez-González et al. [65], who show that healthy subjects do not experience reduction in tissue SpO${}_{2}$, even though the blood flow may be hindered due to a long-lasting pressure load.

Our interpretation, the inverse relationship between SpO${}_{2}$ levels and PU risk, is also supported by the correlations observed between patients with previous PU surgery and the measured SpO${}_{2}$ as discussed in Section 3.2: patients who had already undergone a PU surgery show significantly reduced mean SpO${}_{2}$, which is also a strong indicator for the correlation of tissue SpO${}_{2}$ and PU risk. This observation aligns well with previous work, e.g., by Yafi et al. [66], who found a similar correlation between a decrease in tissue oxygen saturation and PU formations in different stages. They used an external imaging system to derive the tissue SpO${}_{2}$, which is not comfortable for ambulant patients.

Nevertheless, there remain certain overarching challenges associated with mobile pulse oximetry measurements (photoplethysmography), which we want to highlight. We cannot exclude all external disturbances in a real-life study. Kim et al. [67] provide a comprehensive analysis of photoplethysmography interference, particularly for wearable devices. Those measurements can be disturbed by patients’ motion or light scattering of the sensor [67]. Since, we are focusing on lying patients, we assume that motion-related artifacts are rare but light scattering might be relevant. Those artifacts should be reduced in the patient group compared to the control group, as patients are extensively monitored. This includes the impact of different skin types, especially for lower SpO${}_{2}$ levels [46,47]. However, light scattering may be also impacted by a varying sensor pressing force [67] that could differ depending on whether patients within the KIPRODE study lie on the sensor or not. We recommend further optimization of the KIPRODE system, with a focus on developing a sensor case that considers the potential variations in applied pressure. Some of those interferences can be reduced by adjusted filtering and intelligent algorithms. For example, Lauterbach et al. [37], present accurate and reliable pulse oximeter measurements with a state-of-the-art wearable device. The same holds for Poorzagar et al. [68], who also reported accurate SpO${}_{2}$ results for poorly perfused patient tissue with modern wearable sensors. The findings of these two studies support our approach of utilizing tissue SpO${}_{2}$ levels as a predictive factor for PUs, with the potential to further enhance signal quality in the near future.

The investigation on how preexisting conditions impact the body vital signs reveals another finding: Section 3.2 has shown a significant decrease in the mean skin temperature for patients who already underwent a PU surgery. These results are in accordance with related research such as Jiang et al. [55] or Rapp et al. [56], who present comparable results with a nursing facility study, showing differences in skin temperature for high and low PU risk patients. In conclusion, substantial evidence indicates a direct correlation between lower skin temperature and compromised tissue perfusion, which increases the risk for PU development. Nevertheless, neither the existing research nor our study has established a direct short-term correlation between skin temperature and PU development, which makes it difficult to provide an ad-hoc PU predictor. It can be helpful as a general PU risk estimation of patients. The SpO${}_{2}$ trends seem to be far more effective for short term PU prediction due to their direct correlation.

As a result of the KIPRODE study, we propose a PU predictor that continuously analyzes SpO${}_{2}$ trends of risk patients at known risk body sites. If it detects a certain pattern within the sensor trend, an alarm is triggered to alert the patient, enabling them to relieve the pressure load. A promising pattern seems to be a rapid decline in tissue SpO${}_{2}$: a reduction of around 7% in 1.5 h. This observation is consistent with the findings of Harris et al. [69], who determine significant tissue injury after three hours and almost 50% loss of vessel reflow after five hours of pressure loading. However, given the limited number of samples who developed a PU, that predictor probably requires further optimization. The following Table 2 summarizes the presented results from our study relative to other relevant research.

## 5. Conclusions

The presented KIPRODE study reveals several interesting findings about tissue blood perfusion, skin health and the risk for developing a pressure ulcer (PU). We designed, built and thoroughly tested a new wearable sensor system: the KIPRODE sensor system. It enables the recording of vital signs at the hip or back (trochanter or sacral) of patients and combines them with the corresponding pressure load measurements in local proximity. We observed a significant decrease in tissue blood oxygenation (SpO${}_{2}$) in the hours (0–4 h) before a PU becomes visible and is detected by a physician. Although we were able to observe only a single instance of PU development due to effective prophylaxis at the study institutes, this observation aligns closely with our originally formulated hypothesis: disturbed tissue perfusion leads to insufficient oxygen supply and thereby reduces skin health. The disturbance in this case arises from prolonged pressure loads experienced by at-risk patients who have limited mobility or are unable to perceive the necessity for repositioning. The collected data permits the proposal of a PU development predictor utilizing machine learning, but requires validation in subsequent research.

Furthermore, we explored additional hypotheses regarding pressure ulcers at patients feet: how does the distribution of pressure loads on patients’ feet during their daily life influence the risk of PU development? Additionally, how can orthopedic shoes aid in foot ulcer prophylaxis by mitigating pressure spikes? Individually adjusted shoes or insoles have been found to significantly decrease both the average pressure load and the occurrence of pressure spikes. Therefore, they offer valuable support for preventing foot ulcers.

The KIPRODE study provides a further step towards understanding the development of PUs and, consequently, their possible prevention. The sensor systems are ready for further studies with larger patient groups to verify the PU predictor and optimize its machine learning algorithms. Establishing a reliable PU prophylaxis would be a critical step in addressing a severe and preventable disease that affects numerous patients.

## Figures and Tables

**Figure 1 bioengineering-10-01125-f001:**
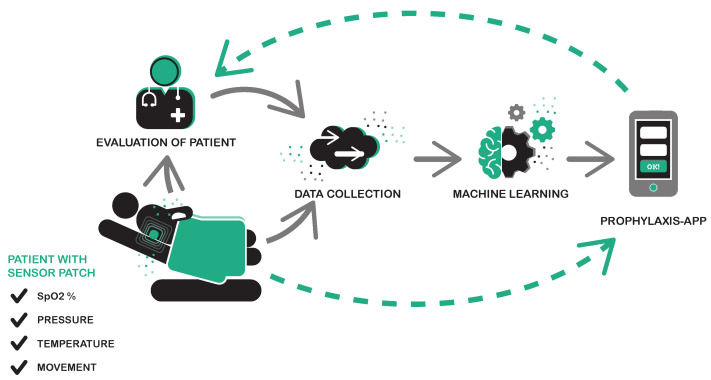
Concept of our suggested pressure ulcer prophylaxis system: KIPRODE [2].

**Figure 2 bioengineering-10-01125-f002:**
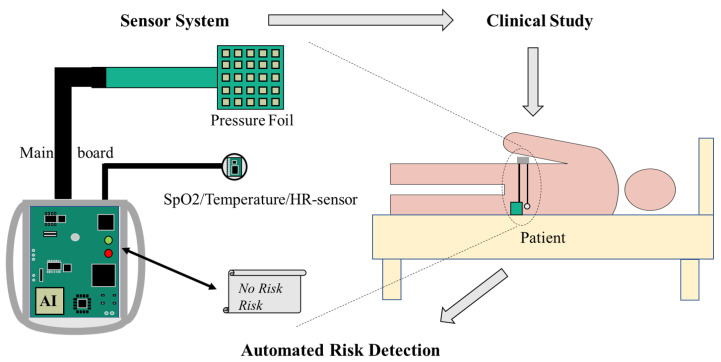
Schematic application of our KIPRODE-system as introduced in [40].

**Figure 3 bioengineering-10-01125-f003:**
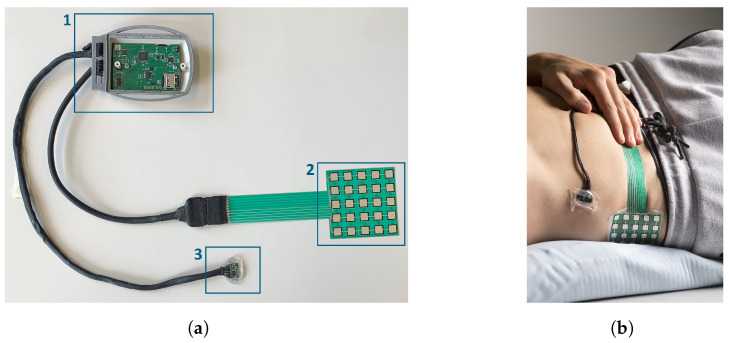
KIPRODE sensor system and its application at a control group patient: the sensor node and pressure sensing foil are located in close proximity to each other at a predilection site. (**a**) KIPRODE sensor system: (1) main electronics: battery powered data acquisition (2) pressure sensor foil (3) sensor node: reflective pulse oximeter and infrared thermometer. (**b**) Sensor applied at a patient: the hip (trochanter) is a predilection site for PUs due to the thin skin over the bones. (^©^Fraunhofer EMFT/Bernd Mueller).

**Figure 4 bioengineering-10-01125-f004:**
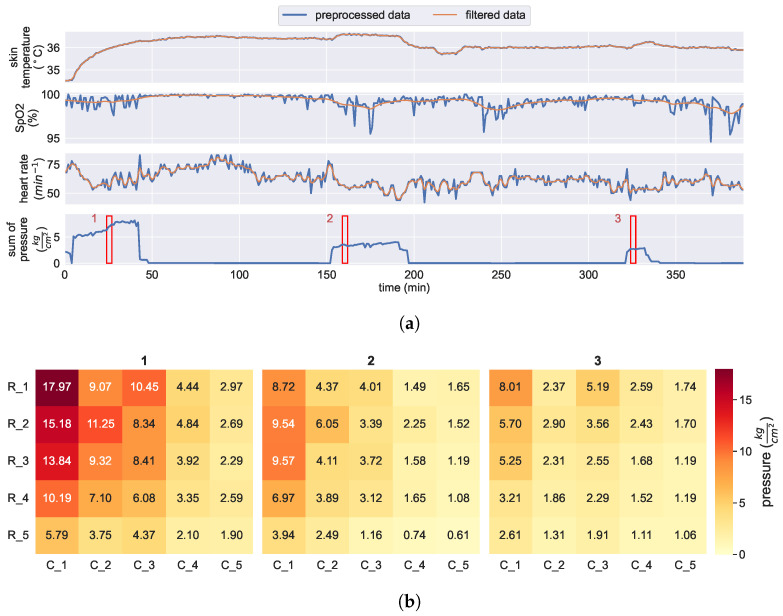
Sensor data acquired by the KIPRODE system. Shown here is the recording of one night of a control group member. (**a**) Vital parameters and sum of pressure data; the latter represents the total pressure load on the sensor foil. (filter parameters: Section A.1); (**b**) pressure foil sensor data acquired at the time point indicated by the red rectangles shown in the Figure above. Each sector represents one of the 25 equidistant sensing elements.

**Figure 5 bioengineering-10-01125-f005:**
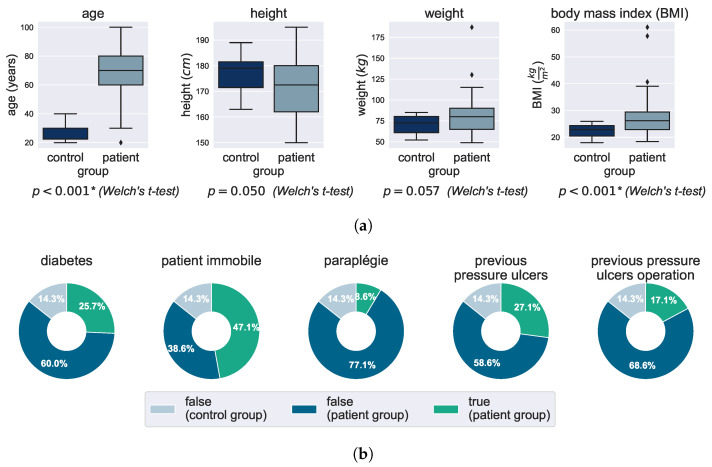
Overview of the study group including patients and control group with regards to their physical data and preexisting conditions. (**a**) Physical data of the control and patient group: the control group is on average younger and in better physical condition. The results of a Welch’s unequal variances *t*-test are reported below each figure, and an asterisk indicates that the differences in means are statistically significant at the level of α=0.001. (**b**) Frequency of preexisting conditions within the study group; the control group (light blue color) is healthy without any known preexisting conditions relevant to the study.

**Figure 6 bioengineering-10-01125-f006:**
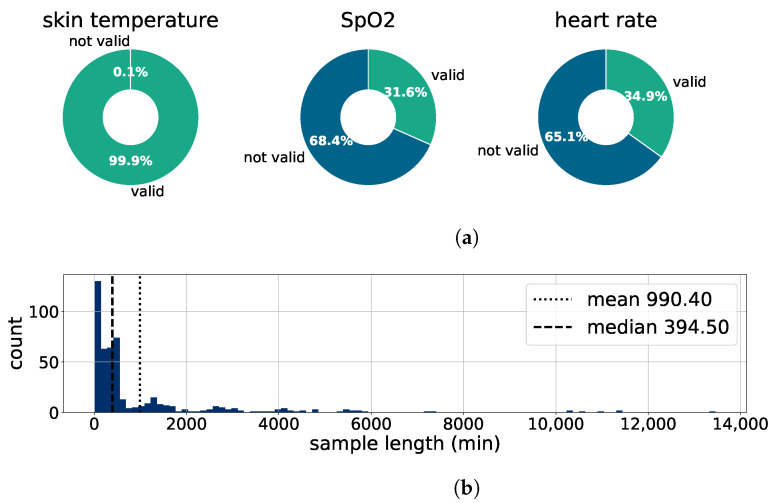
Overview of data quality in the whole dataset with regards to valid measurements. (**a**) Amount of valid values in the preprocessed dataset. (**b**) Distribution of the length of consecutive valid values in the preprocessed dataset.

**Figure 7 bioengineering-10-01125-f007:**
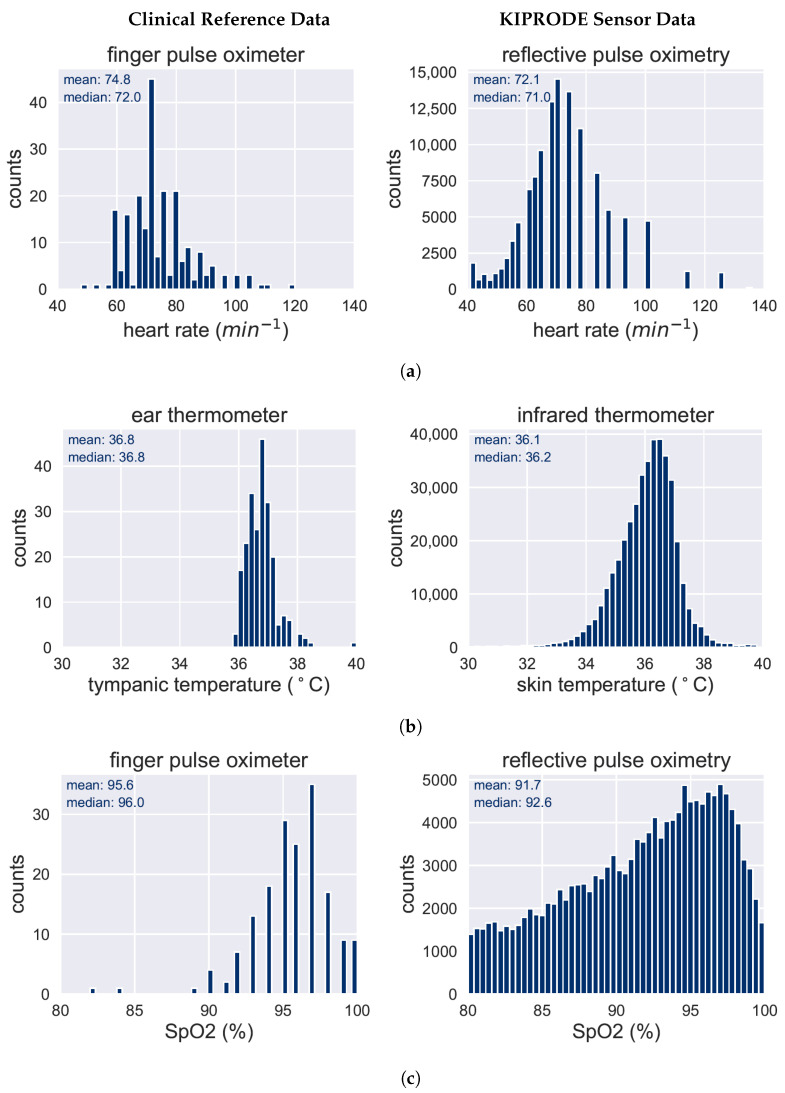
Comparison of the recorded vital signs measurements with our newly developed KIPRODE sensor system with common-practice medical reference data. The latter is plotted on the left and only determined a few times a day, whereas the KIPRODE system on the right measures with a one-minute sampling rate. (**a**) Comparison of heart rate (HR) measurements: similar distribution in both datasets. (**b**) Temperature measurements; note that reference and sensor data show different body sites. (**c**) Distribution of SpO${}_{2}$ readings, again measured at different body sites: noticeable left-shift of the distribution for KIPRODE sensor data on the right.

**Figure 8 bioengineering-10-01125-f008:**
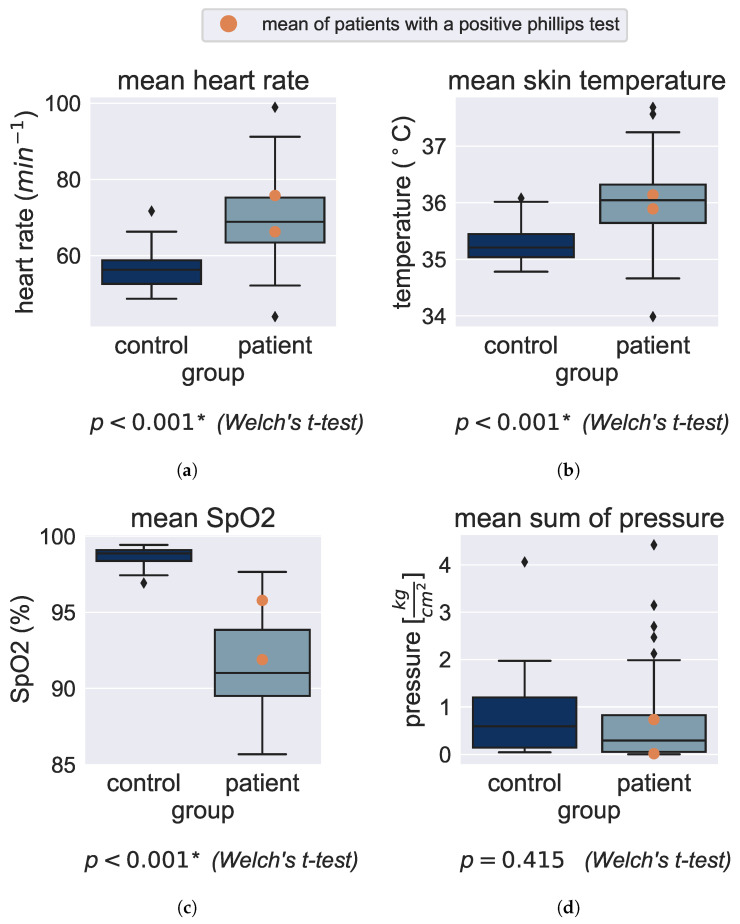
Comparison of recorded vital signs of healthy subjects with the patient group: results of the control group reside in expected ranges, whereas patients show significant deviations. (**a**) Mean heart rate (HR): as expected, we see a relevant increase for patients; on average they are older and several have preexisting conditions (see Section 2.4). (**b**) Skin temperature: increased temperature for patients. The difference could be due to a higher ambient temperature in a hospital, so the analysis focuses on relative changes. (**c**) Blood oxygen saturation, SpO${}_{2}$: strongly decreased median of mean SpO${}_{2}$ for patient group suggesting worse tissue blood circulation. (**d**) The mean pressure load for patients and control group is similar, thus, it does not explain the existing differences in (**a**–**c**). The results of a Welch’s unequal variances *t*-test are reported below each figure, and an asterisk indicates that the differences in means are statistically significant at the level of α=0.001.

**Figure 9 bioengineering-10-01125-f009:**
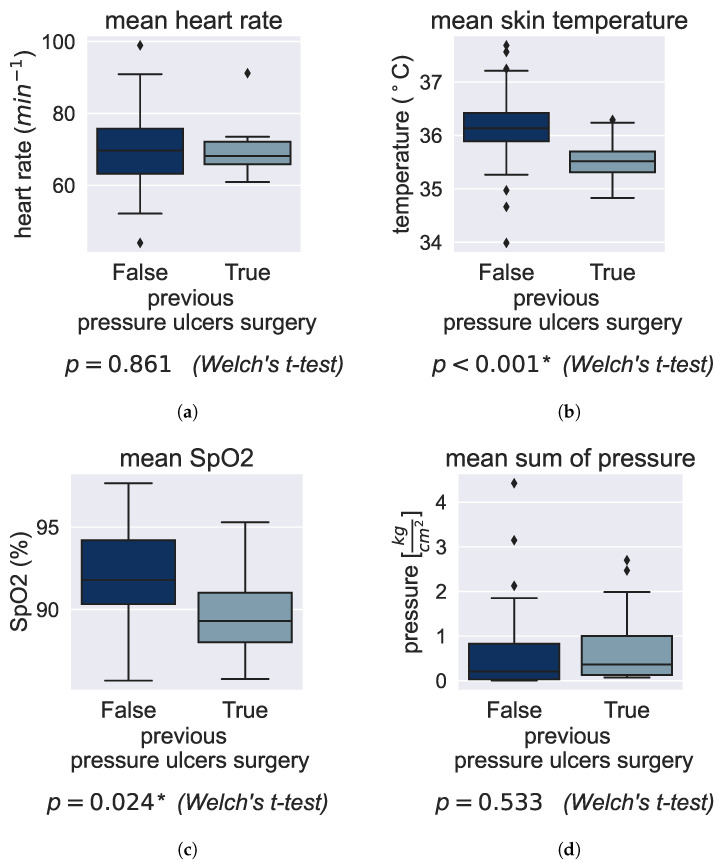
Influence of a single preexisting condition within the patient group: here patients with and without a PU surgery prior to the study are compared (for other preexisting conditions, see Section A.3). (**a**) The mean heart rate does not depend on a previous PU surgery. (**b**) A reduced mean skin temperature in patients with a previous surgery, which could be an indicator of deteriorating tissue perfusion. (**c**) The tissue blood oxygen saturation, SpO${}_{2}$, is significantly reduced for patients with this precondition, suggesting decreased tissue blood circulation. (**d**) The mean pressure load for patients with or without a previous PU surgery is quite similar, hence, the pressure load cannot explain differences in (**b**,**c**). The results of a Welch’s unequal variances *t*-test are reported below each figure, and an asterisk indicates that the differences in means are statistically significant at the level of α=0.001.

**Figure 10 bioengineering-10-01125-f010:**
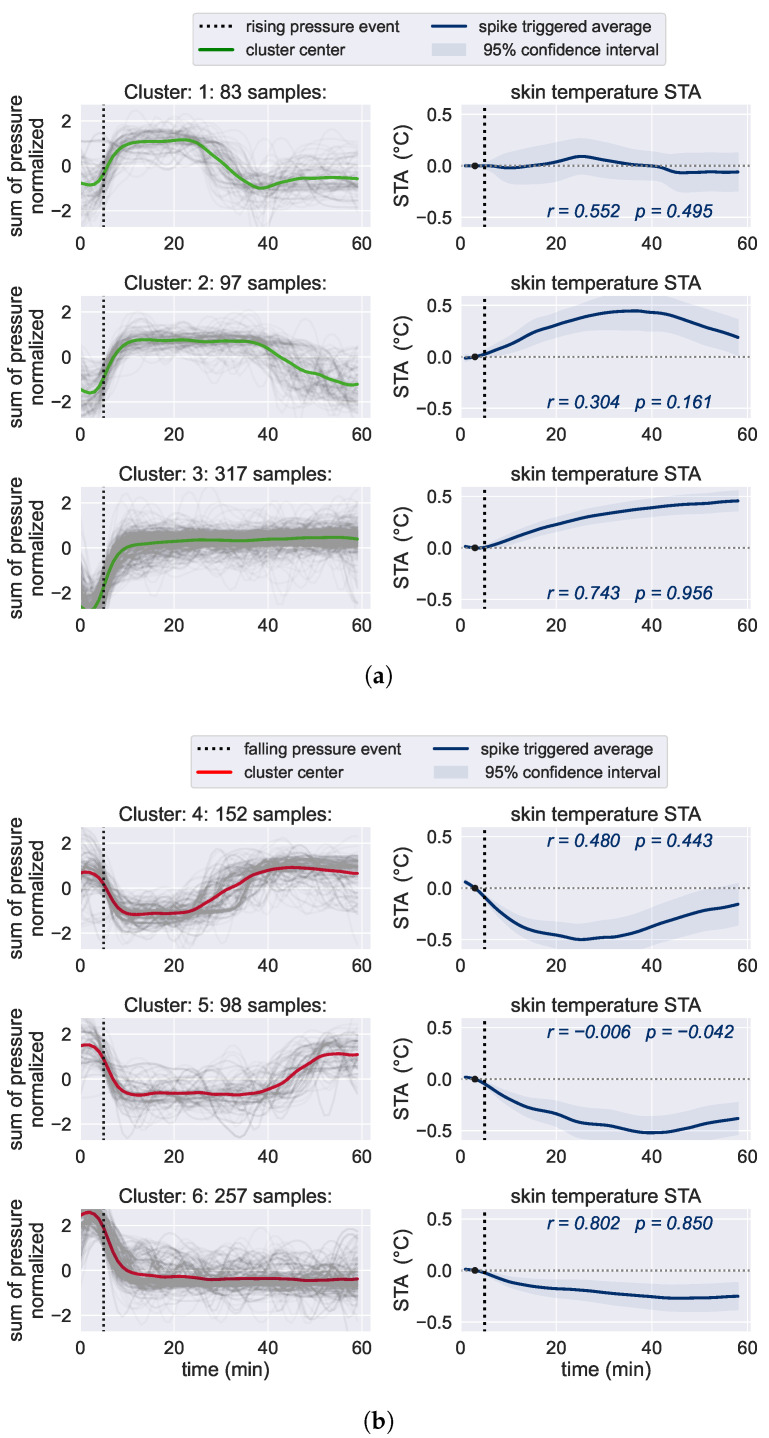
Skin temperature reaction with respect to pressure loads: for every cluster of similar pressure loads, the corresponding spike triggered average of the skin temperature is shown. The correlation coefficients according to Pearson (*r*) and Spearman (ρ) were calculated between the pressure cluster center and the spike triggered average [54]. (**a**) Rising pressure events. (**b**) Falling pressure events.

**Figure 11 bioengineering-10-01125-f011:**
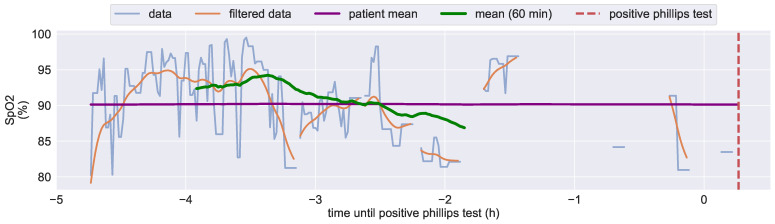
The tissue blood oxygenation measured at the predilection site six hours before a positive Phillips finger test diagnoses a stage one PU. The patient mean is the expanding mean of patient data available at that timestamp. (filter parameters: Section A.1).

**Figure 12 bioengineering-10-01125-f012:**
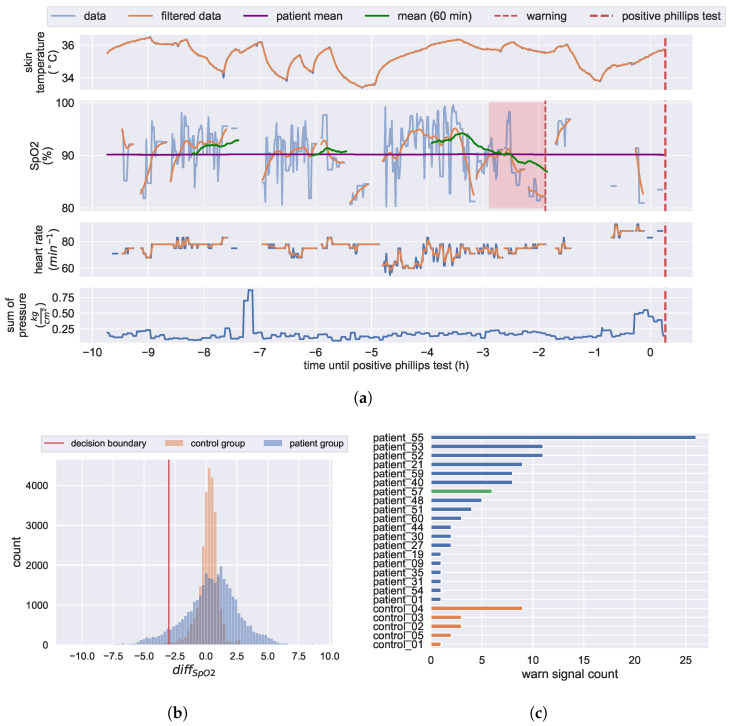
The proposed PU predictor as described in Equation (Equation 1). As only two positive Phillips finger test were recorded, we analyze how often the predictor would raise a warning in the control and in the patient group. (**a**) Vital signs and pressure measurements during the 10 h before a positive Philips finger test was recorded. In the SpO${}_{2}$ subfigure it is visualized how a warning is raised. If the difference between the rolling mean of the last hour and the expanding mean of the patient data available at that time falls below the threshold shown in (**b**) a warning is raised. (filter parameters: Section A.1). (**b**) Distribution of differences between the rolling means of SpO${}_{2}$ over the last hour and the patient data available at that timestamp as defined in Equation (Equation 1). (**c**) Number of warnings the proposed predictor raises per patient. Highlighted in green is the patient whose data is shown in (**a**).

**Figure 13 bioengineering-10-01125-f013:**
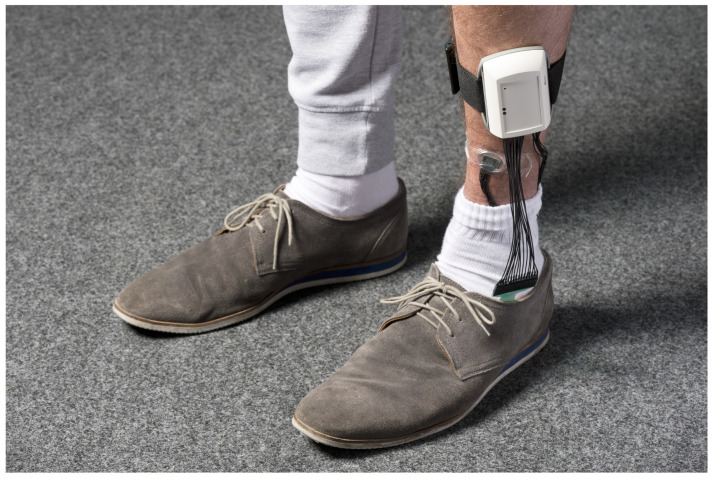
Adapted KIPRODE sensor system to investigate the pressure load of risk patients at their feet during their daily life. The white box contains all electronics, whereas the small sensor node measures vital signs. The pressure sensor foils is inserted in the shoe (^©^Fraunhofer EMF/Bernd Mueller).

**Figure 14 bioengineering-10-01125-f014:**
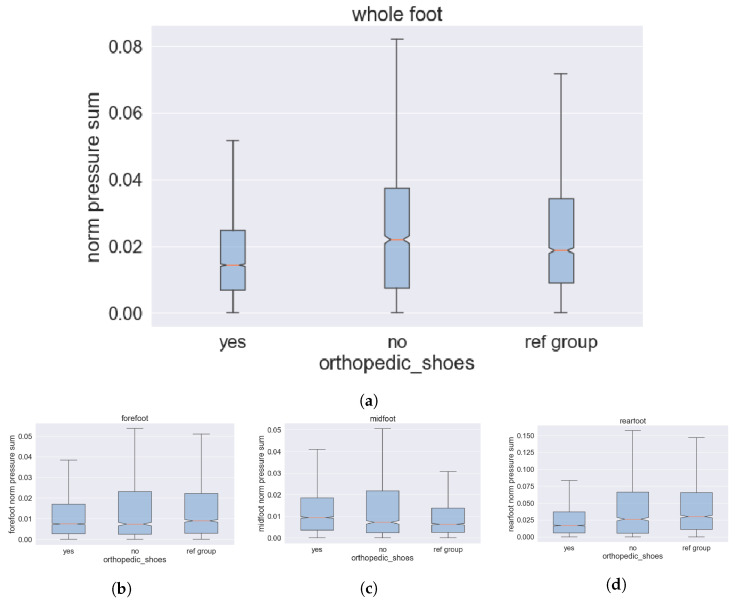
The graph displays the summarized pressure load on patients’ feet, normalized to their weight. The sole of the feet is split into three sub-areas: forefoot, midfoot and rearfoot. All areas display notably reduced pressure loads for patients wearing orthopedic shoes. (**a**) Normalized feet pressure load: the left bar represents patients with individually optimized orthopedic shoes, the middle bar represents patients without such shoes, and the right bar represents the healthy control group. (**b**) Normalized pressure load at the forefoot. (**c**) Normalized pressure load at the midfoot. (**d**) Normalized pressure load at the rearfoot/heel.

**Table 1 bioengineering-10-01125-t001:** Study population at the MRI with corresponding length of participation.

Study Duration	Patient Group	Control Group
Total	60	10
More than 10 days	12	10
More than 5 days	29	10
More than 3 days	44	10

**Table 2 bioengineering-10-01125-t002:** Comparison of our findings with the literature.

Research	Reference	Methods	Findings	Analogy
Tissue SpO${}_{2}$ during long-lasting pressure load	Gómez-González et al. [65]	Doppler laser devices and infrared beams for determining the degree of oxygenation	Healthy subjects do not experience a reduction in tissue SpO${}_{2}$	Agrees well with Section 3.1.2
Assess skin status in care	Yafi et al. [66]	Near-infrared Spatial Frequency Domain Imaging (SFDI)	Decrease in tissue oxygen saturation due to PUs	Aligns well with Section 3.4.1
Quality of pulse oximeter in wearables	Poorzagar et al. [68]	Literature review	Oximeters, particularly newer models, are accurate in poorly perfused patients	Confirms the KIPRODE measurement method
Effects of skin pigmentation on the accuracy of SpO${}_{2}$ measurements	Shi et al. [46]	Literature review	Pulse oximetry overestimates SpO${}_{2}$ of dark-skinned people	Must be considered in future design
Effects of skin pigmentation on the accuracy of SpO${}_{2}$ measurements	Cabanas et al. [47]	Literature review	Pulse oximeters are less accurate in dark-skinned individuals at lower SpO${}_{2}$	Must be considered in future design
Validate the skin temperature on sacral region as early warning signs of PU	Jiang et al. [55]	Relative skin temperature at sacral area measured daily	Differences in skin temperature for patients with high/low PU risk	Aligns well with Section 3.2
Influence of skin temperature regulation on PU risk	Rapp et al. [56]	Skin temperature monitors	PU risk depends on multiscale entropy for skin temperature	Aligns well with Section 3.2
Effect of varying duration of ischemia on tissue damage	Harris et al. [69]	Tissue blood analysis in animals	Changes in blood composition after lasting ischemia	Aligns well with Section 3.4.1

## Data Availability

To enhance the research for PU prophylaxis, we provide our data on request at MediaTUM https://mediatum.ub.tum.de/1717646 (accessed on 13 August 2023) [41] and our code on Fraunhofer GitLab https://gitlab.cc-asp.fraunhofer.de/mls/public (accessed on 13 August 2023) [42].

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
