# Peer review of "Wearable Prophylaxis Tool for AI-Driven Identification of Early Warning Patterns of Pressure Ulcers"

_bioengineering, 2023, doi:10.3390/bioengineering10101125_

Round 1

Reviewer 1 Report

Manuscript ID: Bioengineering-2579808

Title: Wearable Prophylaxis Tool for AI-driven Identification of Early Warning Patterns of Pressure Ulcers

Author: Gruenerbel et al.

In this manuscript, authors have developed a pressure ulcer prophylaxis device (PU sensor system) which is able to monitor the pressure load and tissue vital signs  in immediate local proximity at patient’s predilection sites.

After carefully reviewing the manuscript, I found that this work is original, and all results are clear and well presented. However, I would like to accept the manuscript but after answering the following minor revision:

1- Fabrication and specification details of  KIPRODE-system should be provided in the manuscript within the experimental section.

2- Results of this study for detecting pressure ulcer should be compared with other reported results published elsewhere. I suggest authors to create a comparison table in the manuscript to highlight their results with other reported results including measurement method.

Minor editing of English language required

Reviewer 2 Report

Review of Manuscript Submission bioengineering-2579808

I have carefully reviewed the manuscript titled "Wearable Prophylaxis Tool for AI-driven Identification of Early Warning Patterns of Pressure Ulcers" submitted to Bioengineering for consideration. This article built a new sensor system able to monitor the pressure load and tissue vital signs in immediate local proximity at patients’ predilection sites. Moreover, propose a prophylaxis system that allows for predicting PU developments in the early stages before they become visible. I think this work is meaningful and can provide a basis and direction for future research. It can be accepted after revision.

1.       The basic principle of blood oxygen level detection is the absorbance of the same photodiode under different blood oxygen concentrations. Therefore, people with different skin tones (different melanin levels) may have different sensitivity to blood oxygen concentration. It was noted that the authors entered several training parameters of other patients, including age, height, weight, and BMI, but without taking skin color into account. Thus, I think this parameter is meaningful for the construction of an AI-driven early warning model for pressure ulcers.

2.       It is noted that Figure.4A shows the filtered data. What is the filter range? Did it the real-time and automatical filtered process, or the post-processed data? If it is post-processed data, how to achieve multi-channel synchronous early warning in practical applications?

3.       The system adjustment for prophylaxis of foot pressure ulcers reveals the potential of practical clinic applications. Does the author take into account the wireless transmission of multichannel signals? For example, add an NFC module. I think this should be meaningful for realizing portable real-time monitoring.

Round 2

Reviewer 2 Report

I am satisfied with authors’ responding, and it can be published in present form.